

# Annual Characterization of Regional Hydrological Drought using Auxiliary Information under Global Warming Scenario

Zulfiqar Ali[1,2], Ijaz Hussain[1], Muhammad Faisal[3,4]

[1]Department of Statistics, Quaid-i-Azam University, Islamabad, Pakistan.
[2]Johann Bernoulli Institute (JBI) Rijksuniversiteit Groningen, Groningen The Netherlands.
[3]Faculty of Health Studies, University of Bradford, BD7 1DP Bradford, UK; Bradford.
[4]Institute for Health Research, Bradford Teaching Hospitals NHS Foundation Trust, Bradford, UK.

*Correspondence to*: Zulfiqar Ali (z.a.ali@rug.nl)

**Abstract.** Climate change and global warming scenario is likely to increase worsening drought across the World. Drought is
a complex natural hazard, which is a composition of many factors such as hydrological, meteorological and agricultural.
Accurate characterization of hydrological drought at regional level is challenging. Standardized Drought Indices (SDI) is
commonly used method for drought characterization and monitoring. In this study, we proposed a hydrological drought index,
which uses improved monthly precipitation estimates under global warming scenario. As monthly precipitation records have
significant role in regional drought characterization. Therefore, this research suggests auxiliary information as local weights
to improve monthly precipitation records in terms of dependence characteristic of temperature with precipitation records under
regression estimation settings. Consequently, we proposed a new method of hydrological drought assessment-the Locally
Weighted Standardized Precipitation Index (LWSDI). We assessed hydrological drought using LWSDI on ten meteorological
stations located in various climatological regions of Pakistan. We compared and evaluated the performance of LWSDI with
Standardized Precipitation Index (SPI) and Standardized Evapotranspiration Index (SPEI) at 12-month time scale based on
Pearson correlation. We found high positive correlation between the LWSDI and existing methods (SPI and SPEI). In
summary, improved estimates of precipitation can strengthen drought-monitoring system.

## 1 Introduction

Climate change and global warming scenario increases the odds of worsening drought in many parts of the world. Drought is
a complex natural hazard, which is a composition of many factors such as hydrological, meteorological and agricultural. Its
complex structure has adverse effect on a wide range of socioeconomic infrastructure (Dracup et al., 1980). Drought can be
defined as "an extended period of deficient rainfall relative to the multi-year mean for a region" (Schneider et al., 2011). There
are four drought types: meteorological, hydrological, agricultural and hydrologic, socio-economic drought. All these are highly
influenced by lack of precipitation amount and its frequency. However, hydrological drought occurs when dry weather patterns
outweigh other climatic conditions.

Lack of precipitation, change in climate, human activities and overexploitation of surface water resources are the main factor
involved in hydrological drought (Van Loon, and Van Lanen, 2013). Further, changes in climatic conditions (Dawadi and
Ahmad, 2013) and contamination of water are the key significant factors which greatly contribute in water scarcity and drought
(Aswathanarayana, 2001). Consequently, perpetual increase in global warming are continuously affecting most regions all
over the world. Especially, in dry regions, droughts will expand and water level will drop, expanding semi-arid to arid regions
(Huang et al., 2016). In addition, several other environmental factors are involved in drought occurrence such as, high wind,



low relative humidity, temperature, characteristics and duration of rain, intensity and onset (Wilhite, 1994), however, the role of long-term precipitation records greatly contributes in all these methods.

There are several drought assessment and monitoring tools, however, Standardized Drought Indices (SDI) are one of the most commonly used procedure around the world (Svoboda et al. 2016). These standardized methods are helpful in making effective
drought monitoring and mitigation policies by adopting early warming strategies. In all SDI methods, long-term temporal precipitation records are prominently contributing factor.

As, the regional patterns of long term rainfall and temperature play a very important role in the continuous monitoring of climate change (Coffel and Horton, 2015) and other natural hazards. Further, the precipitation records and its regional dissemination is the key responsibility of meteorological division and weather monitoring department. Therefore, it is
necessary for drought management to define and monitor drought conditions using more representatively regional data and procedures. Consequently, accurate quantification and precise estimates of rainfall are required for widespread ranges of research in meteorology, hydrology, atmospheric and other disciplines. However, in developing countries, it is difficult to implement standard climatic monitoring guidelines such as Canadian Avalanche Association, (1995) and Egan and Baldelli, (2009) (Overpeck, 2011).

In recent years, optimized rain gauge monitoring networks and their implementation have great importance in the estimation procedure of rainfall (Chebbi et al., 2013). There are several studies that suggest optimized meteorological network for measuring precipitation records. Hong et al., (2016) proposed a GIS-based technique to establish optimal metro-hydrological station network of Vu Gia- Thu Bon river basin. Adhikary et al., (2015) proposed an optimal gauging network for measuring precipitation for Middle Yarra River catchment of Australia. Chacon-Hurtado et al., (2017) reviewed some existing application
and monitoring network and proposed an optimal network for modeling hydrological data. Other related research includes (Amorim et al., 2012); (Kurdzo and Palmer, 2012); (Sandar et al., 2015); (Baume, 2011). However, adoptions of these technique, raises several questions regarding the accuracy in rainfall estimation (Yoon and Lee, 2017). Firstly, the rainfall is a spatial climatic variable that vary both in spatial and temporal dimensions. Secondly, its short distance variability characteristics and being a spatial variable suggest complex structure of the optimized meteorological network (Scarsoglio et
al., 2013; Einfalt et al., 1990). However, the high cost of installation and complex sampling design may force to adopt compromise allocation and installation of meteorological station with respect to all environmental variables.

As the distribution of rainfall and extreme temperature are the main factors of the watershed (Courty et al., 2018), therefore, these two climatic variables have great impact and substantial importance in advance hydrological research. Consequently, to infer the complexity in regional climate and its temporal structure, several authors jointly configured these two environmental
processes by using advanced statistical models and geospatial tools. Some of them are (Coles and Tawn, 1991); (Guler et al., 2007); (Mahdian et al., 2009). However, these methods are based on temporal data recorded at single stations, that is, it only covers a single realization at continuous spatial domain. This deprived the study and findings from the spatial prevalence effect of climate nature and climate change. Further, the increases in uncertainty in the estimation may lead bad impact on climate change policies and reliability of forecasting environmental characteristics. Therefore, the efficiency and regional
representativeness of regional precipitation records integrated with temperature at recording stage may seek to provide significant contribution in accurate and reliable drought mitigation policies.

However, recent developments in precise and accurate estimation methods that incorporate auxiliary are available in sampling theory and estimation. Cochran, (2007) provides detailed theory and methods related the use of auxiliary information to improve the estimation of unknown characteristics of random variables. In environmental science and ecological statistics,
several authors integrate auxiliary information to improve the estimation of study variable under auxiliary information based statistical methods such as regression and kriging (ZHU & Lin, 2010). Paloscia et al., (2013) used Landsat images, which are





close to the ENVISAT data to observe the effect of the vegetation. Apaydin et al., (2011) used elevation as a source of auxiliary information for the interpolation of climatological variables under co-kriging settings.

To keep the importance of auxiliary variable in the estimation process, this research aimed to integrate and utilize coherence structure of regional behavior of temperature as an auxiliary variable. We provide a new framework and rationale of using temperature as an auxiliary information for precise estimation of mean precipitation. The estimated values of mean precipitation are further used to obtain Standardized Drought Indices (SDI) values. Unlike, precipitation, temperature can be seen as a global representative at specific natural homogenous optimized regional catchment area. Therefore, to improve annual precipitation record by using temperature as an auxiliary variable is straightforward and rationally valid.

This article gives a new procedure of annual drought characterization in which improved precipitation records are configured in SDI procedure. Here, to reduce the sampling error in the estimated quantity of recorded precipitation amount and to account effect of global warming in drought monitoring, we suggest giving local weights in term of dependence characteristic of monthly records of temperature to precipitation records under regression estimation settings. From the result of sampling and estimation theories, we see that, when there is positive correlation between study and auxiliary variable, regression estimator incorporating auxiliary information play very important role to enhance the representative characteristics of estimates.
Consequently, this paper proposed a new method of hydrological drought assessment the locally Weighted Standardized Precipitation Index (LWSDI). LWSDI used locally weighted improved estimates of precipitation in the standardized procedure of SDI. Application of the proposed methods is made for ten meteorological stations located in various climatological regions of Pakistan. Here, we collect time series data of monthly total precipitation and mean monthly temperature from 1971 to 20017 from Pakistan meteorological department of Pakistan. For comparative analysis, we include Pearson correlation statistic and
two existing procures of SDI that are Standardized Precipitation Index (SPI) and Standardized Evapotranspiration Index (SPEI) at 12-month time scale.

For computation of LWSDI, SPI and SPEI, this research employed various probability distribution criterion provided by Stagg et al., (2015) in SDI procedure. In addition, the study advocate optimal non-parametric methods for handling uncertainty related to extreme event. In recent research, a non-parametric approach – a graphical method is suggested as a substitute of
probability distributions (Hao, et al., 2014). As the probability plotting position which is very closed to parametric distribution have more capability to address the extreme events with greater efficiency, therefore, in the presence of extreme values in the data, accurate and precise analysis are expected under non-parametric approach.

## 2 Material and Methods

### 2.1 Data and Study Area

In this study, we consider 10 meteorological stations that are scattered in all the climatic locations in Pakistan. Pakistan have a multiple geographical diversity simultaneously and characterized by the annual cycle of climate variability. Figure 1 shows the locations of selected meteorological stations of Pakistan. These stations are highly variable in rainfall and temperature during different seasons. Pakistan contains remarkable landscapes as it stretches from Arabian Sea, its southern border, to the world's magnificent mountain ranges in the north (Qamar-uz-Zaman et al., 2009). The drought has become a recurrent
phenomenon in the country. In every season, some of the stations are continued to bear extremely vulnerable drought condition. In the recent decade, due to severe drought hazard, the economic system of the country badly disturbed, besides of several human life and livestock waste. Most of the country is arid to semi-arid except southern slopes of Himalayas and Sub Mountain region where the annual rainfall ranges from 760mm to 2000mm. Three-fourth part of the country receives rainfall less than 250mm and 20 percent of it receives 125mm. Pakistan has four well marked seasons: Cold, from November to February;
Premonsoon (Hot), from March to mid of June; Monsoon, from mid of June to mid of September; Post-monsoon, from mid of September to October (Khan, 1993). Summer season is extremely hot and the relative humidity ranges from 25 percent to 50 percent. The major part of Pakistan is arid to semi-arid with large spatial variability in the temperature (Chaudhry, 2009). In



this research monthly total precipitation and mean temperature data (January 1970 to December 2017 is collected from Pakistan Meteorological Department. Table 1 shows the general statistics of monthly data ranges from 1970 to 2017 at selected stations.

## 2.2 Regression Estimation: Auxiliary Information Based Efficient Estimation

As there are many optimization networks to record meteorological observations such as precipitation and temperature (Amorim et al., 2012; Kurdzo and Palmer, 2012; Chebbi et al., 2013). However, the choice of their implementation depends on the geographical structure of regions and the financial resource (Mishra and Coulibaly, 2009). Moreover, particularly in developing countries, it is very hard to assess and report these observations in an optimized way (Parker, 2017). Precipitation records are based on a single realization of specific stations that is, it only constitutes a sample from a continuum that cannot be recorded everywhere (Webster and Oliver, 2007). Thus, the estimated characteristics of precipitation such as mean monthly total must have certain statistical properties for example unbiasedness and minimum variation. However, precipitation being an environmental variable has certain notable characteristics. It has huge regional variability, that is, in monsoon season, substantial variation in precipitation amount and its occurrence are observed at a short distance (Rafiuddin et al 2010). So, before defining regional climatology and the temporal deficient in rainfall, it is necessary to improve temporal precipitation records. Here, advance statistical procedure may be incorporated to increase the validity and regional representativeness of precipitation amount under spatial setting.

Thus, to improve sample characteristics of precipitation records, there are several estimation methods incorporating auxiliary information in the idea phase. These methods include ratio, product and regression estimation (Tripathi et al., 1994; Rao, 1988). Even so, depending upon the nature of data and sampling strategy, each method has certain conditions. However, regression estimation is one of the important and the most excessively used technique in survey sampling. In many surveys and records keeping modules, a collection of some extra information related to study variables are common practice. It provides estimates that are more efficient when the study variable is a positively correlated with auxiliary variables and then incorporation of regression estimation (Royall and Cumberland, 1981; Isaki and Fuller, 1982; Singh and Espejo, 2003). In recent years, several studies have developed various estimators that utilized auxiliary variable under regression estimation settings. Recently, Lu (2013) constructs a new chain regression estimator which utilized two auxiliary variables. His theoretical results proven that the proposed estimators are more efficient than the traditional regression estimators. Awan and Shabbir, (2014) proposed, a difference-in-regression estimator by using two auxiliary variables in simple random sampling. It is tested by both mathematically and numerically that, when there is a positive correlation between study variable and auxiliary variables then the regression estimator gives more representative and efficient estimates. In earlier research, several studies used regression estimator (Marker, 1999; Dong et al., 2017; Van, 2013; Li, 2007 and several others).

The general expression of the simple regression estimator is as follows (Cochran, 2007).
$$\overline{y}_r = \overline{y} + b(X - \overline{x}) \qquad (1)$$

Where, $\overline{y}_r$ is the update (regression) mean of study variable, $\overline{y}$ is the sample mean study variable, $X$ is the overall mean auxiliary variable, $\overline{x}$ is the sample of auxiliary variable and $b$ is the regression slop.

We consider the problem of updating and improving regional precipitation estimates using auxiliary information under global warming scenarios. We propose temperature as an auxiliary information for improving the precipitation records. Since temperature being a globally representative environmental variable and homogenous in nature has a strong association between precipitation, the use of temperature as an auxiliary data is logically valid. Various surveys indicate that there is a positive correlation between rain and temperature. Zhao and Khalil (1993) examine the relationship between precipitation and temperature in all the seasons for eight regions including the USA. At the Guliya ice core, detailed analyses of the precipitation index (glacier accumulation) and the temperature proxy recorded since 300 years BP show that precipitation correlates with temperature in this region (Yang et al., 2006). Rajeevan et al., (1998) found that temperature and rainfall were positively correlated during January and May but negatively correlated during July at India. Sneva (1997) found positive month-wise correlation between temperature and rainfall in southeastern Oregon. Other recent studies supporting a positive correlation



between temperature and precipitation in various climatic regions can be found in (Jain et al., 2013; Sun and Du, 2017;Chen et al., 2013;Mueller and Seneviratne, 2012).

This study suggests temperature as an auxiliary information to improve annual meteorological records of precipitation after
assessing theoretical support about the positive correlation between precipitation and rainfall. So, before defining drought characteristics and precipitation deficient, we utilized the concept of regression estimator to improve the annual estimates of precipitation using temperature as an auxiliary information. However, if there is zero correlation or negative, the research advocates another technique such as ratio and product methods (Singh, and Espejo, 2003)

The mathematical structure of regression estimator employing monthly mean temperature as an auxiliary information for the
estimation of total monthly amount of precipitation is as follows.

$$LW = \overline{r} + b(T - \overline{t})  \qquad (2)$$

Where, LW are the improved precipitation estimates of mean, where, simple precipitation records are weighted by the dependence characteristics of temperature with precipitation. $\overline{r}$ is the annually moving mean of simple precipitation records, T is the overall mean of study station and $\overline{t}$ is the annually moving mean of simple mean monthly temperature records.

**2.3 Proposed new hydrological drought index – Locally Weighted Standardized Drought Index (LWSDI)**

In this paper, instead of using simple precipitation records, we suggest LW estimates of precipitation estimates for drought assessment. As LW estimates are more representative and account direct effect of temperature in estimation stage, therefore, the preference of using of LW estimate instead of simple records is logically accepted. Hence, to obtain SDI, following of Stagg et al., (2015), this study incorporates the standardization of those Cumulative Distribution Function (CDF) of various
probability distribution that fits on temporal series of LW estimates. In addition, the study includes non-parametric methods for assessing the validity of the proposed study. A brief description on both methods are as follows.

**1) Parametric methods: Selection of appropriate probability distributions**

In this method, a list of candidate distributions among several existing probability functions is prepared. Therefore, for appropriate selection, we proposed several available probability distributions such as Gamma, Normal, generalized Pareto and
Log-Normal to check their candidacy for modeling LW time series at individual meteorological station.
In numerical evaluation and screening of appropriate probability distribution, goodness of fit techniques includes Chi-square, Anderson darling and Kolmogorov Samirnov test. However, we suggests multi-parameter probability distributions to make a list of candidate distributions. For example, instead of two-parameter Gamma and two parameter Weibull, goodness of fit should be applied on the three-parameter Gamma and four-parameter Weibull, respectively.
Further, those probability functions that have the lowest value of Bayesian Information Criteria (BIC) are used to obtain SDI. Thus, the CDF of the selected distribution is transformed by the following methods,

$$G(x) = q + (1 - q)F(x)  \qquad (3)$$

In Equation 3, a little modification in the CDF is made to adjust the effect undefined values. For example, in case of Gamma distribution $q$ is the probability of zero value in the time series data LW. If $m$ be the number of zeros presented in LW time
series data, then $q$ is estimated by $m/n$, where $n$ is the total number of observation in the LW time series.


### 2) Non-Parametric Methods: Integration of graphical techniques

As, in each probabilistic models, uncertainty always exists in accurate and precise estimation (Parker, 2014). Moreover, the selection of probability distributions for each indicator is purely subjective in nature. To avoid these problems, Hao and AghaKouchak (2014) proposed an idea of non-parametric for drought monitoring. They used Gingorten probability position
formula (Gringorten, 1963) as an alternative of Gamma distribution for obtaining standardized drought index. The fact of using graphical method is to capture the uncertainty of extreme events and hence to reduce errors in accurate and precise estimation of drought indices. Several other studies used non-parametric approach for drought monitoring (Ghamghami et al., 2017; Farahmand and AghaKouchak, 2015; Zhang et al., 2017). However, use of only one probability plotting position is not enough for data of varying behavior. As only Gamma distribution is not enough to capture several regional behavior in the procedure
of drought indices (Stagg et al., 2015). Therefore, the utilization of various probabilies plotting formula are required to assess the behavior of specified probability distribution (Shukri Yah et al., 2012; Vogel, 1986).
In this study, we used five most commonly used probability plotting position for non-parametric evaluation of SDI and computation of LWSDI. Table 2 provides the list of proposed probability plotting methods.

*Standardization*
This step requires appropriate transformation methods to standardize the selected cumulative density function, and all the numerical vectors comprising the time series data based on the probability plotting formulas. Therefore, following the same procedure of SPI and SPEI (Farahmand and AghaKouchak, 2015; Ghamghami et al., 2017), standardized data on both (parametric and non-parametric) versions of SDI is obtained by the following equations (Abramowitz and Stegun, 1965),

$$LWSDI = -\left( z + \frac{c_o + c_1 z + c_2 z^2}{1 + d_1 z + d_2 z^2 + d_3 z^3} \right) \qquad (4)$$

for

$$z = \sqrt{\ln\left[ \frac{1}{\{T(x)\}^2} \right]}$$

when

$$0 < T(x) \le 0.5 \qquad (5)$$

$$LWSDI = +\left( z - \frac{c_o + c_1 z + c_3 z^2}{1 + d_1 z + d_2 z^2 + d_3 z^3} \right) \qquad (6)$$


and for,

$$z = \sqrt{\ln\left[ \frac{1}{\{1 - T(x)\}^2} \right]}$$

when,

$$0.5 < T(x) \le 1 \qquad (7)$$

where,

$$T(X) = \begin{cases} G(X) & \text{iff to standardized the fitted parameteric distributions} \\ P(X_i) & \text{iff to standardized probability plotting posstion formulas,} \end{cases}$$

$c_0=2.515517$, $c_1=0.802853$, $c_2=0.010328$, $d_1=1.432788$, $d_2=0.985269$, $d_3=0.001308$ are constant factors.

## 2.4 Comparative Statistics and Methods





The study includes Pearson Product-Moment Correlation coefficient, commonly termed as the correlation coefficient (r) for comparing the outcomes of the proposed methods with existing methods. Correlation coefficient (r) is a measure of co-linearity between two arrays and most widely used test statistics.
It is computed using Eq. (1).

$$r = \frac{\sum_{i=1}^{n}(x-\overline{x})(y-\overline{y})}{\sqrt{\sum_{i=1}^{n}(x-x)^2 \sum_{i=1}^{n}(y-\overline{y})^2}} \qquad (8)$$

where, $x_i$ and $y_i$ represent the values of arrays with 'n' number of elements being compared and, $\overline{x}$ and $\overline{y}$ are the mean values of two arrays. The range of "r" lies between -1 to 1. Positive values near to 1 shows strong correlation between the two arrays. While negative values show inverse relationship.

In hydrological research, especially in the evaluation of new droughts and comparative study of drought indices, the correlation coefficient is most commonly used statistical method (Vicente-Serrano et al., 2010; Ali et al., 2017; Tsakiris, 2005).
As there are several multi-scalar drought indices which can be used to report drought severity and drought intensity. However, in this research, we assessed the performance LWSDI with SPI and SPEI at 12 the month time scale. The selection of these indices is due their analogous standardized mathematical procedure and classification criterion. Moreover, these indices are
the most commonly used and worldwide accepted methods. A brief description of these indices are as follows.

*SPI index*

McKee et al. (1993) developed an SPI drought index, which is based on long term precipitation record to quantify the precipitation scarcity for different time scales at the single monitoring station. Initially, SPI method is based on the standardization of CDF of the gamma distribution. Where negative and positive SPI values designate less than or greater than
median precipitation, respectively (Bordi and Sutera, 2007).
However, in latter research, it has been shown that the use of only gamma distribution is inadequate to model rainfall data (Blain et al., 2015; Stagge et al., 2015). Therefore, a more appropriate probability distribution function is required among the list of available probability functions. In estimation process, this research follows the guidelines of Stagge et al., (2015). we compute SPI index at the 12 month time scale for all the regions. The general expression of SPI index can be written as,

$$P_i \sim pdf \,(\text{Paramter 1, Parameter 2,...., Parameter } n)$$

where $P_i$ is the monthly cumulative total of rainfall, and $pdf$ showing the optimal probability function with "$n$" parameters. Estimation of quantitative values of SPI can be made by normalizing the CDF of appropriate selected probability distributions which are fitted on the observed monthly cumulative precipitation time series.

*SPEI Index*

Analogous to SPI, Vicente-Serrano et al., (2010) developed a new multi-scalar drought index: the Standardized Precipitation Evaptranspiration Index (SPEI). In SPEI, water balance model based on the difference between precipitation and Potential Evapotranspiration (PET) is suggested in the same methodological framework of SPI. Symbolically,

$$D_i \sim pdf \,(\text{Paramter 1, Parameter 2,...., Parameter } n)$$

where, $D_i$ is the water deficient indicator and is defined as,

$$D_i = P_i - PET_i ,$$

here, $P_i$ and $PET_i$ are the monthly total precipitation and potential evapotranspiration respectively.

Recent applications of SPEI based regional studies of drought are available in the literature (Hui-Mean et al., 2018;Ma et al., 2015;Miah et al., 2017; Das et al., 2016). In this research, we employed SPEI (Beguería et al., 2013) package for the estimation





of PET values. However, the rest procedure of calculating SPEI values at the 12 month time scale is same as recommended by  Stagge et al., (2015).

## 1.    Results and Discussions
### 3.1  Temporal behaviour and deviations

Drought characterization based on a 12-month timescale (annual) provides an overall picture of regional hydrological conditions (Gumus and Algin, 2017). Here, the SPI-12 is the most commonly used drought index for annual monitoring and characterization of hydrological drought (Habibi et al., 2017). In previous research, several studies proposed hydrological drought indices and compared it with SPI-12 (Cumbie-Ward et al., 2016; Jain et al., 2015; Naumann et al., 2014; Xuchun et al., 2016). In this study, before the standardization of LW, a little graphical analysis is done by assessing the temporal behavior
and deviations in improving precipitation records with those which are used in the SPI-12 index. SPI-12 uses twelve-month moving average of monthly precipitation records. Analogous to SPI, the proposed structure of precipitation records used in the LW model have the same mathematical structure and rationale. Therefore, it is necessary to show how auxiliary information play role in the temporal estimation of precipitation records. In this study, we randomly took four stations for graphical representation of temporal precipitation records. Figure 2 shows the temporal and density plots of both locally weighted and
usual records of precipitation at Sialkot, Sargodha, Jehlum and Pasni stations. We found significant changes, especially in upper records (i.e above averages) of Sialkot, Sargodha, and Jhelum. Further, the density plots also significantly deviate from each other. However, a small amount of difference is observed at Pasni station. These differences show how drought analysis and characterization of regional climatology can be amended by introducing advanced techniques of estimation.  Further, before standardization, we may find the various proposal of the best probability distribution of simple precipitation and
improved precipitation records.

### 3.2  Parametric computation of drought indices and comparisons.

To compute and assess the comparative evaluation of LWSDI based drought indices, we use the monthly time series secondary data on total precipitation and average temperature from 1971 to 2017.  Parallel to LWSDI the study also computes SPI and SPEI. Therefore, for each indicator, a list of highly parameter probability distributions is prepared to check their fitness. In this
study we assess 32 probability distributions by using *Propagate* (Spiess, 2014) an R package. Here we used various fitness criterion such as Kolmogorov-Smirnov, and Chi-Square, Anderson-Darling tests were used to find the candidate distribution for each indicator. For each stations, the CDF of those distributions, which have minimum value of BIC, are then selected for standardization.
Table 3 provides CDF of selected probability functions in all indicator at all study stations.  Selection of each probability
distribution is based on (weighted) residual sum-of-squares as the minimization criterion based on Levenberg-Marquardt algorithm (Moré, 1978). Where, the estimation phase consists on method of moments, method of maximum likelihood estimation and method of L-moments. All these methods were implemented using *lmom* Hosking and Hosking (2017) R packages. Table 3 provides the parameters of optimal distribution parameters used to obtain LWSDI, SPI and SPEI for all the stations. Especially in SPI and LWSDI, fitness and selection of different probability distribution validate the finding of (Stagg
et al., 2015).
Figure 3 presents the temporal behavior of LWSDI, SPI-12 and SPEI-12 index. In all station, the temporal behavior of LWSDI strongly correlated with SPI-12. However, a little deviation is observed between SPEI. Table 4 provides numerical assessment about the correlation of LWSDI with SPI-12 and SPEI-12. In all stations, the correlation between LWSDI and SPI is significantly high. Irrespective to low correlation (0.46) of SPEI-12 with LWSDI in Quetta, the other stations have strong
correlations. This shows that LWSDI can be recommended as an alternate drought index which incorporate auxiliary information based precipitation data for characterization of hydrological drought.

### 3.3  Non-Parametric evaluation and comparisons

Beside of careful selection of probability model, our experimental findings show that extreme values and outlier are beyond the coverage of the appropriate probability functions.
Figure 4 shows that the exponential distribution with lowest BIC among all the available probability models, does not capture the uncertainty in the significant parts of the data for both SPI and LWSDI model in Pasni station. Moreover, substantial deviation can be seen in Gamma and Normal model at Jhelum stations.  Similar results were found for other stations and distribution.  All selected distribution seemed to ineffective for reporting extreme observations. Therefore, to check and



validate more, the study integrates five probability plotting position formula. After standardization the data of probability plotting based vector of time series, we see that LWSDI is remain aligned with SPI and SPEI. In all probability plotting function, the correlation of LWSDI with SPI and SPEI is significantly high in all station except Sialkot (See Table 5). In Sialkot, all the nonparametric functions show weak correlation of LWSDI with SPI and SPEI.

Consequently, to obtain SDI series, it is necessary to consider highly parameter probability distributions. However, in practice it is very difficult to include all the parametric distribution in drought monitoring protocol. Alternative to parametric version of drought indices, this paper suggests graphical techniques of probability plotting positions. However, the selection of appropriate probability plotting formula is important to capture the uncertainty related upper and lower extreme events as well.

## 2.    Conclusions

This study strengths drought-monitoring module by integration of improved precipitation records in standard guidelines of SDI procedure. Where, improvements of precipitation records are based on the utilization of auxiliary information at the estimation stage of mean annual rainfall amount. Here, the simple regression estimator method is employed to weight simple moving average of annual rainfall amount. Consequently, the study proposed a new hydrological drought index: The Locally Weighted Standardized Drought Index (LWSDI). Efficiency of LWSDI is assessed by checking the relationship among the
quantitative values of SPI and SPEI under Pearson correlation statistics.

Figure 1 shows experimental findings of 10 meteorological stations that are located in various climatological regions of Pakistan. We used time series data of monthly total rainfall and mea monthly temperature from 1971 to 2017. To compute the SDI under LWSDI, SPI and SPEI, methodology of estimation consists on both parametric and non-parametric approach. Here varying distribution concept (Stagg et al., 2015) is employed for parametric standardization. Moreover, the study
extended the idea of non-parametric standardization (Hao and AghaKouchak, 2014) by including five probability plotting methods (See Table 2).

Following conclusions can be drawn from the study,

1.    It is noted that there is a significant difference between the distributions (See Figure 2; Table 4) of simple records and of those, which are improved by including the mean temperature as an auxiliary information under regression
estimator.
2.    Instead of varying distribution-based standardization, LWSDI gives closed estimated values to SPI and SPEI (See Figure 3).
3.    Overall, comparative analysis shows that LWSDI is strongly correlated with SPI under both parametric and non-parametric standardization. However, some discrepancies are found with SPEI in parametric standardization (See
Table 4).
4.    In non-parametric standardization, LWSDI is strongly correlated with both SPI and SPI in most of the study region. However, weak but positive correlation exist only in Sialkot station (See Table 5).
5.    Above mentioned findings shows that the direct role of temperature at the estimation stage of rainfall makes configurations of using LWSDI based drought analysis multivariate drought indices settings.

The key advantage of the proposed index is to monitor hydrological drought using more regionally improved precipitation data.

However, the limitations of the study are 1) the proposed method of drought monitoring have lack of multi-scaling characteristics. 2) for particular stations, where the relationship between precipitation and temperature is negative and the hypothetical supportive features of regression seems to fail, LWSDI is not best a choice. Here some advance techniques of
estimation may incorporate to justify the relationship between precipitation and temperature such as ration and product estimator.

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



Figure 1: Study Area





Table 1: Climatology of the selected study regions

| Station Name | Latitude | Longitude | Precipitation Multi-annual | | | Temperature Multi-annual | | |
|---|---|---|---|---|---|---|---|---|
| | | | Mean | S.D | C.V | Mean | S.D | C.V |
| Sargodha | 72.67111 | 32.08361 | 40.05 | 55.610 | 138.84 | 24.4524 | 7.783 | 31.831 |
| Gupis | 73.24000 | 36.10000 | 15.87 | 29.982 | 188.97 | 12.638 | 8.722 | 69.010 |
| Nawabshah | 68.40955 | 26.24833 | 12.21 | 37.117 | 304.12 | 26.891 | 7.065 | 26.273 |
| Jehlum | 73.72637 | 32.93313 | 73.84 | 94.872 | 128.48 | 23.715 | 7.158 | 30.185 |
| Karachi | 67.08220 | 24.90560 | 15.92 | 41.540 | 260.99 | 26.612 | 4.443 | 16.694 |
| Pasni | 63.41540 | 25.25100 | 8.5 | 23.371 | 276.18 | 25.793 | 4.2882 | 16.626 |
| Quetta | 67.00971 | 30.19900 | 21.7 | 35.673 | 164.29 | 16.557 | 8.523 | 51.474 |
| Sialkot | 74.53134 | 32.49268 | 84.04 | 128.98 | 153.47 | 22.943 | 7.195 | 31.360 |
| Drosh | 71.80380 | 35.56840 | 47.25 | 46.099 | 97.561 | 17.631 | 9.066 | 51.418 |
| Jiwani | 61.77070 | 25.05380 | 8.8 | 26.380 | 300.73 | 25.775 | 4.023 | 15.609 |

Table 2: Probability Plotting Position formulas

| List | Proponent (Authors) | Expression |
|---|---|---|
| PP-1 | Hazzan Allen (1914) | $P(x_i) = \dfrac{2i-1}{2n}$ |
| PP-2 | Gringorten Gringorten (1963) | $P(x_i) = \dfrac{i-0.44}{n+12}$ |
| PP-3 | Tukey Tukey (1962) | $P(x_i) = \dfrac{i-0.333}{n+0.333}$ |
| PP-4 | Weibull Weibull (1939) | $P(x_i) = \dfrac{i}{n+1}$ |
| PP-5 | Laplace Lund et al. (1995) | $P(x_i) = \dfrac{i-1}{n+2}$ |




Figure 2: Deviation between simple and improved precipitation records

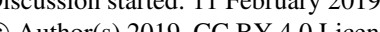



Figure 4: Observed Probability Distributions

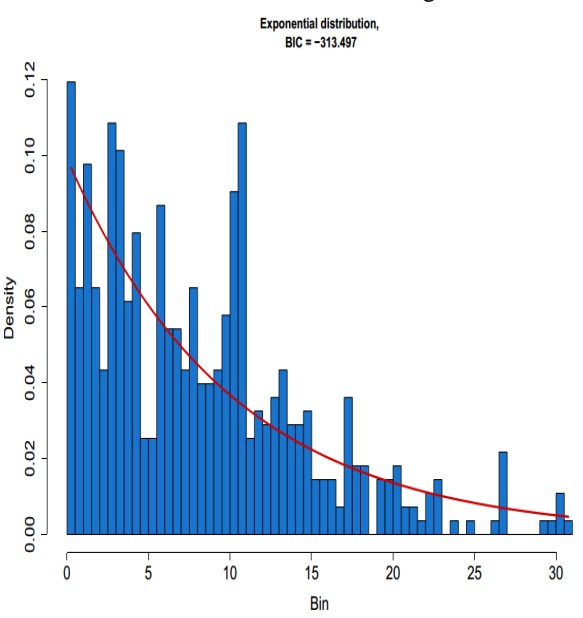

Simple Precipitation Records (Pasni)

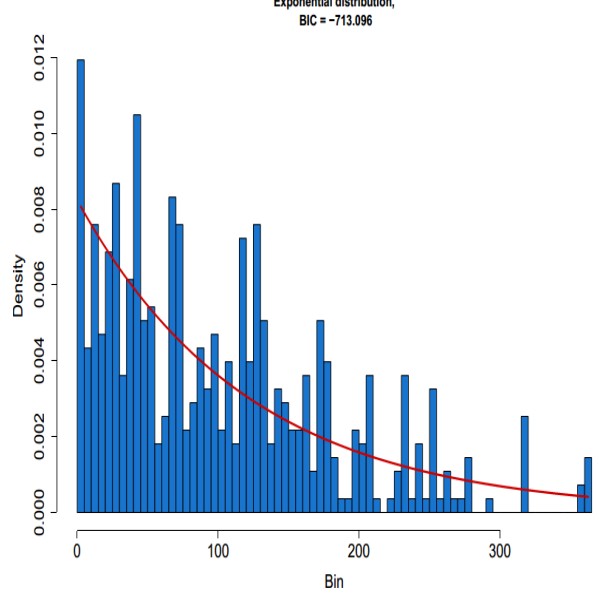

LW based Improved Precipitation Records (Pasni)

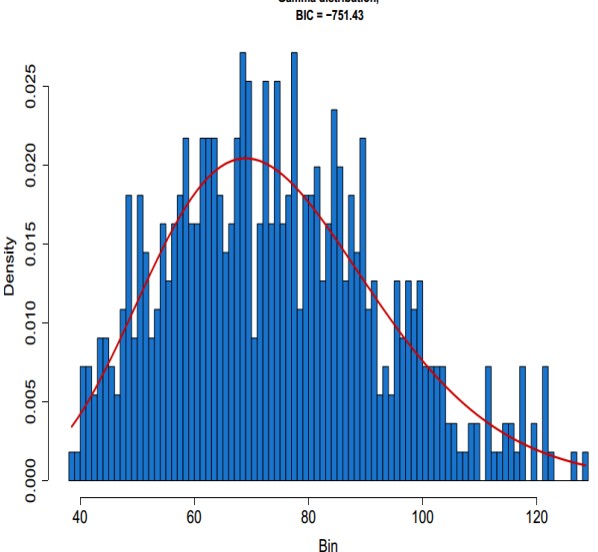

Simple Precipitation Records (Jehlam)

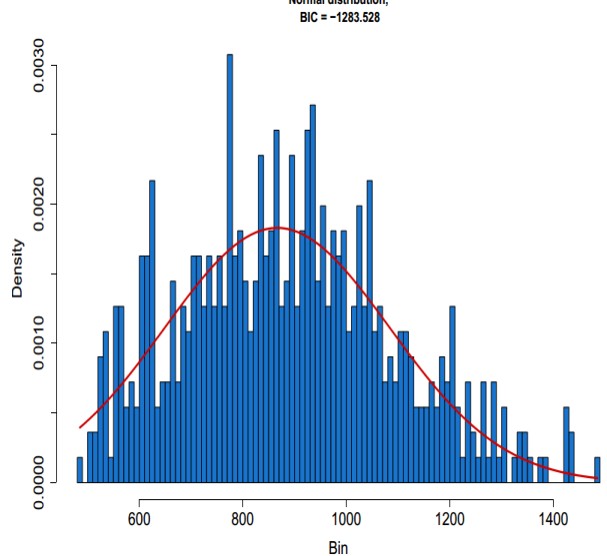

Simple Precipitation Records (Jehlam)





Table 3: Optimal distributions and their parameters

| Stations | Statistics | LWSDI | SPI | SPEI |
|---|---|---|---|---|
| Sialkot | Function | Gumbel | Gumbel | Gumbel |
| | Parameters | $\mu=70.81, \beta=23.26$ | $\mu=863.26, \beta=264.76$ | $\mu=-710.93, \beta=302.49$ |
| | BIC | -1116.20 | -1155.49 | -1168.23 |
| Jhelum | Function | Gamma | Normal | Normal |
| | Parameters | $k=13.64, \theta=0.18$ | $\mu=866.411, \sigma=217.95$ | $\mu=777.236, \sigma=260.725$ |
| | BIC | -751.4299 | -1283.528 | -1555.059 |
| Sargodha | Function | Gumbel | Gamma | Normal |
| | Parameters | $\mu=34.01, \beta=12.06$ | $k=8.10, \theta=0.01$ | $\mu=-1247.38, \sigma=193.72$ |
| | BIC | -815.3444 | -935.6124 | -1161.51 |
| Pasni | Function | Exponential | Exponential | Normal |
| | Parameters | $\lambda = 0.09915$ | $\lambda = 0.00824$ | $\mu=-1491.95, \sigma=118.77$ |
| | BIC | -313.4968 | -713.096 | -1340.081 |
| Nawab Shah | Function | Exponential | Exponential | Laplace |
| | Parameters | $\lambda = 0.07612$ | $\lambda = 0.00629$ | $\mu=-2010.46, b=161.81$ |
| | BIC | -608.8158 | -1398.404 | -989.389 |
| Karachi | Function | Exponential | Gamma | Gumbel |
| | Parameters | $\lambda = 0.04961$ | $k=1.23, \theta=0.0057$ | $\mu=-1478.10, \beta=132.78$ |
| | BIC | -561.6398 | -1184.320 | -1386.529 |
| Quetta | Function | Laplace | Logistic | Normal |
| | Parameters | $\mu=19.53, b=11.67$ | $\mu=234.01, \sigma=62.50$ | $\mu=-1363.08, \sigma=133.89$ |
| | BIC | -812.869 | -1067.321 | -1409.171 |
| Gupis | Function | Laplace | Cuachy | Laplace |
| | Parameters | $\mu=9.28, b=9.39$ | $X_o=116.82, \Upsilon=54.90$ | $\mu=-1068.86, b=145.27$ |
| | BIC | -852.131 | -933.833 | -1063.664 |
| Drosh | Function | Gamma | Normal | Normal |
| | Parameters | $k=13.56, \theta=0.28$ | $\mu=561.02, \sigma=153.68$ | $\mu=-864.35, \sigma=195.02$ |
| | BIC | -696.0621 | -1536.87 | -1191.154 |
| Jiwani | Function | Exponential | Gamma | Gumbel |
| | Parameters | $\lambda = 0.09849$ | $k=1.08, \theta=0.0094$ | $\mu=-1382.99, \beta=91.38$ |
| | BIC | -326.2558 | -735.703 | -1517.208 |



Figure 3: Temporal representation of LWSDI, SPI and SPEI



Table 4: Parametric evaluation- Correlation of LWSDI with SPI-12 and SPEI-12

| Stations | SPI | SPEI |
|---|---|---|
| Gupis | 0.916051 | 0.916051 |
| Drosh | 0.977754 | 0.957310 |
| Pasni | 0.994351 | 0.856361 |
| Jiwani | 0.989041 | 0.823774 |
| Quetta | 0.965612 | 0.464298 |
| Nawab Shah | 0.998303 | 0.849928 |
| Karachi | 0.991821 | 0.941572 |
| Sialkot | 0.993306 | 0.986141 |
| Jehlam | 0.977865 | 0.979398 |
| Sargodha | 0.991214 | 0.935256 |

Table 5: Non-parametric evaluation: Correlation of LWSDI with SPI-12 and SPEI-12

| | District | Hazzan | Weibull | Tukey | Laplace | Gringorten |
|---|---|---|---|---|---|---|
| **SPI** | Gupis | 0.997792 | 0.997786 | 0.997790 | 0.997788 | 0.997791 |
| | Drosh | 0.980751 | 0.980735 | 0.980742 | 0.980834 | 0.980747 |
| | Pasni | 0.992276 | 0.992276 | 0.992276 | 0.992264 | 0.992276 |
| | Jiwani | 0.991150 | 0.991141 | 0.991146 | 0.991138 | 0.991148 |
| | Quetta | 0.978905 | 0.978807 | 0.978869 | 0.978720 | 0.978891 |
| | Nawab Shah | 0.997854 | 0.997936 | 0.997892 | 0.997973 | 0.997870 |
| | Karachi | 0.992830 | 0.992908 | 0.992856 | 0.992895 | 0.992840 |
| | Sialkot | 0.638734 | 0.640137 | 0.639277 | 0.639500 | 0.638943 |
| | Jehlam | 0.990651 | 0.990895 | 0.990741 | 0.991144 | 0.990685 |
| | Sargodha | 0.963490 | 0.963887 | 0.963662 | 0.964160 | 0.963559 |
| **SPEI** | Gupis | 0.884507 | 0.884008 | 0.884340 | 0.884232 | 0.884447 |
| | Drosh | 0.952826 | 0.952870 | 0.952833 | 0.953143 | 0.952827 |
| | Pasni | 0.878759 | 0.878853 | 0.878827 | 0.879126 | 0.878790 |
| | Jiwani | 0.851462 | 0.850687 | 0.851173 | 0.850876 | 0.851353 |
| | Quetta | 0.933514 | 0.933420 | 0.933505 | 0.933332 | 0.933515 |
| | Nawab Shah | 0.817960 | 0.820587 | 0.818980 | 0.824855 | 0.818351 |
| | Karachi | 0.944990 | 0.946326 | 0.945526 | 0.946348 | 0.945199 |
| | **Sialkot** | **0.664168** | **0.665738** | **0.664778** | **0.665247** | **0.664403** |
| | Jehlam | 0.988161 | 0.988789 | 0.988432 | 0.989336 | 0.988269 |
| | Sargodha | 0.938316 | 0.938662 | 0.938466 | 0.938744 | 0.938376 |

