# Peer review of "Annual Characterization of Regional Hydrological Drought using Auxiliary Information under Global Warming Scenario"

_Natural Hazards and Earth System Sciences, 2018_

## Referee Comment (RC1) · Anonymous Referee #1 · 19 Mar 2019

This study proposed an improved method to calculate regional hydrological drought indices by incorporating auxiliary information, i.e., temperature, which is important to investigate hydrological extremes under global warming. However, the intro, results and discussion can be improved. I do have several concerns that should have been addressed before it can be considered for publication. 1. The authors used a new method to calculate hydrological drought index using temperature (LWSDI). LWSDI is not just a drought index, only when LWSDI is smaller than a threshold, it can be used for drought identification. Most importantly, the authors failed to demonstrate the improved LWSDI is more appropriate than SPI/SPEI (or other hydrological drought indices) in investigating hydrological drought event. Although there are good relationships between

[Figure]

LWSDI and SPE/SPEI, it is not convinced to prove LWSDI is better than the existing drought indices. 2. The first two sentences of the ABSRACT and INTRODUCTION are the same. Please rewrite. 3. Line 23-33: The authors failed to explain the relationships among different types of droughts."Hydrological drought occurs when dry weather patterns outweigh other climate conditions". This sentence is rather difficult to understand. Generally, hydrological drought is considered as the water shortage in surface/subsurface water during a certain period.The LWSDI is estimated by precipitation and temperature, which might be a kind of meteorological drought index. 4. Page 4 Line 40: 'various survey indicate that there is a positive correlation between rain and temperature...'.But in following, the authors indicate there are negative correlations between them (Rajeevan et al., 1998). It seems the method proposed by this study only can be used when there is a positive relationship between precipitation and temptation. Therefore, I strongly recommend the authors to give the relationship between them in the manuscript. 5. Before selecting the appropriate probability distributions, I suggest the authors pay more attention to check whether the precipitation time series are stationary before statistical modelling. 6. The results and discussion are rather poor. The title of this study is "annual characterization...".The authors only show the statistics of the employed methods, but don't state the annual characterization of hydrological drought in Pakistan. There might be severe drought in the study area during the study period, it would be better to identify and characterize these droughts, and compare with the other drought indices.
* * *

---

## Referee Comment (RC2) · Anonymous Referee #2 · 19 Mar 2019

NHESS_2018_373 Title: Annual Characterization of Regional Hydrological Drought using Auxiliary Information under Global Warming Scenario Authors : Zulfiqar Ali et al.

GENERAL MAJOR COMMENTS The paper claims to present a hydrological drought index, which uses improved monthly precipitation estimates under global warming scenario. The monthly station precipitation timeseries are improved using local weights utilizing regression equations between precipitation and temperature. Temperature is treated as an auxiliary variable. The paper proposes a new drought index, the Locally Weighted Standardized Precipitation Index (LWSDI) for drought assessment. LWSDI is applied in ten stations across Pakistan for the period 1970-2017 and LWSDI timeseries

are compared at 12-month time scale with the commonly used and well known drought indices SPI and SPEI.

Major Comments

There are many points that should be clarified before considering the paper for publication.

1. The title, abstract and the core of the paper claim that the LWSDI is a hydrological drought index, which is not. However, the time scale of 12-month may suggest that this is a hydrological/water resources drought index. There are many papers that have identify the importance of drought indices time scale in characterizing meteorological and/or hydrological droughts, and the authors should refer to them. 2. The title of the paper (and a few sentences in the Abstract and Introduction) is not appropriate and it does not reflect the research presented. The LWSDI is not a hydrological drought index and the paper does not use an Global Warming Scenario. 3. The various climatic regions of Pakistan should be presented in the "Study Area" section of the paper, since it is claimed that the 10 stations used are representative of these climate regions. 4. It seems that the proposed index is essentially the SPI using locally weighted precipitation. Thus, it is expected that the proposed index LWSDI to compare well with SPI. I suggest that the authors try to analyze specific common and extreme drought events using LWSDI, SPI, SPEI, derive the drought parameters (i.e. drought duration, severity, intensity, etc) for each drought index and compare the results. 5. The authors claim that if a positive linear relationship exists between precipitation and temperature then the proposed method could be applied. However, the relationships for the 10 stations are not presented. These relationships should be presented and thoroughly been discussed, since the methodology is based on these relationships. The discussion of the relationships should be linked with the climatic features of the 10 station locations. 6. The empirical probability distributions of precipitation in the 10 stations should be discussed and the discussion should be linked with the climatology of the regions of Pakistan.

Minor comments 1. There many sentences that need rephrasing. For example: a. Page 2. Line 4. "…(SDI) are one of the most …." Please revise – check English b. Page 2. Line 30. "Some of them are….." Please revise c. Page 3. Line 18. Please correct 20017 to 2017. d. Page 4. Line 10. What is "unbiasedness"??? Please rephrase-correct e. Page 4. Line 28. "…..by both mathematically and numerically…" Please correct. And others.

2. The first two sentences of Abstract and Introduction are the same. Please revise having in mind the major comment #2. 3. Figure 1. Should present the elevation (DEM) of Pakistan. The figure should be a proper map of Pakistan having scale, legend and North symbol. 4. Equation 3. Define F(x) and G(x).

The presented study falls within the scope of NHESS. However, the paper is not ready for publication and needs major revisions before it would be acceptable for publication in the journal of NHESS.

Please also note the supplement to this comment:
https://www.nat-hazards-earth-syst-sci-discuss.net/nhess-2018-373/nhess-2018-373-RC2-supplement.pdf

---

## Author Comment (AC1) · 8 Apr 2019

GENERAL MAJOR COMMENTS The paper claims to present a hydrological drought index, which uses improved monthly precipitation estimates under global warming scenario. The monthly station precipitation timeseries are improved using local weights utilizing regression equations between precipitation and temperature. Temperature is treated as an auxiliary variable. The paper proposes a new drought index, the Locally Weighted Standardized Precipitation Index (LWSDI) for drought assessment. LWSDI is applied in ten stations across Pakistan for the period 1970-2017 and LWSDI timeseries are compared at 12-month time scale with the commonly used and well known drought

indices SPI and SPEI. Major Comments There are many points that should be clarified before considering the paper for publication. 1. The title, abstract and the core of the paper claim that the LWSDI is a hydrological drought index, which is not. However, the time scale of 12-month may suggest that this is a hydrological/water resources drought index. There are many papers that have identify the importance of drought indices time scale in characterizing meteorological and/or hydrological droughts, and the authors should refer to them. Authors Response: We agree with the reviewer. In literature, there is too much confusion between the time scales and the definitions of drought such as meteorological, agricultural and hydrological (Bazrafshan et al., 2014). However, many authors have considered that twelve-month time scale is suitable for defining hydrological drought (Svoboda et al., 2012). In the present work, a statistical methodology is adopted to improve time series data of precipitation. On the same manners of SPI, the proposed data is further used to compute drought indices. The output of the proposed method of data is compatible to compare SPI-12 and SPEI-12. Therefore, we called LWSDI as a hydrological Index, because it gives monthly numerical standardized values at 12-month time scale. References: Bazrafshan, J., Hejabi, S., & Rahimi, J. (2014). Drought monitoring using the multivariate standardized precipitation index (MSPI). Water resources management, 28(4), 1045-1060. Svoboda, M., Hayes, M., & Wood, D. (2012). Standardized precipitation index user guide. World Meteorological Organization Geneva, Switzerland. 2. The title of the paper (and a few sentences in the Abstract and Introduction) is not appropriate and it does not reflect the research presented. The LWSDI is not a hydrological drought index and the paper does not use a Global Warming Scenario. Authors Response: The revised version of the title will be more appropriate and representative. The prospective title is as follows: "Characterization of Regional Hydrological Drought using Improved Precipitation Records under Auxiliary Information"

3. The various climatic regions of Pakistan should be presented in the "Study Area" section of the paper, since it is claimed that the 10 stations used are representative of these climate regions. Authors Response: In the revised version of the manuscript,

we will cite the article of Qasim et al., (2014). This article assesses and provides statistical evidence regarding variation in all parts of the country. In addition, it provides zonal classified map and statistics which are indicating variation among each zone. References: Qasim, M., Khlaid, S., & Shams, D. F. (2014). Spatiotemporal variations and trends in minimum and maximum temperatures of Pakistan. J Appl Environ Biol Sci, 4(8S), 85-93. 4. It seems that the proposed index is essentially the SPI using locally weighted precipitation. Thus, it is expected that the proposed index LWSDI to compare well with SPI. I suggest that the authors try to analyze specific common and extreme drought events using LWSDI, SPI, SPEI, derive the drought parameters (i.e. drought duration, severity, intensity, etc) for each drought index and compare the results. Authors Response: We are thankful to the reviewer for his valuable suggestions. However, inclusion of the suggested analysis and procedures (i.e. drought duration, severity, intensity, etc) are beyond the scope of the objective. 5. The authors claim that if a positive linear relationship exists between precipitation and temperature then the proposed method could be applied. However, the relationships for the 10 stations are not presented. These relationships should be presented and thoroughly been discussed, since the methodology is based on these relationships. The discussion of the relationships should be linked with the climatic features of the 10 station locations. Authors Response: In casestudy experimentation, the correlation between precipitation and temperature is positive. For the revised paper, we have now prepared a table of correlation between precipitation and temperature. In addition, the significance of the correlation is tested by T-test. 6. The empirical probability distributions of precipitation in the 10 stations should be discussed and the discussion should be linked with the climatology of the regions of Pakistan. Authors Response: We have a complete set of analysis data and graphs. We will make a panel graph of the probability distribution. In light of these graphs, the discussion section will be improving accordingly.

Minor comments 1. There many sentences that need rephrasing. For example: a. Page 2. Line 4. ". . .(SDI) are one of the most . . .." Please revise – check English Authors Response: In revised version of the manuscript, we have now revised the

sentence. b. Page 2. Line 30. "Some of them are. . ..." Please revise Authors Response: Thank you reviewer. We have now revised it. c. Page 3. Line 18. Please correct 20017 to 2017. Authors Response: We have now corrected the typo. d. Page 4. Line 10. What is "unbiasedness"??? Please rephrase-correct Authors Response: In statistics and sampling theory, unbiasedness is the property of an estimator. For the best understanding, we will try to rephrase it accordingly. e. Page 4. Line 28. ". . ...by both mathematically and numerically. . ." Please correct. Authors Response: Thank you. The grammatical mistakes and typo will be corrected from native English speaker.

And others. 2. The first two sentences of Abstract and Introduction are the same. Please revise having in mind the major comment #2. Authors Response: In revised version of the manuscript, we now have revised the sentence accordingly.

3. Figure 1. Should present the elevation (DEM) of Pakistan. The figure should be a proper map of Pakistan having scale, legend and North symbol. Authors Response: Here, the map of study area is flexible for its improvement. In a revised map, we will follow the reviewer instructions and suggestions. 4. Equation 3. Define F(x) and G(x). Authors Response: F(x) is the notation of the cumulative distribution function of any probability distribution. While G(x) is the modified cumulative distribution function. In the revised manuscript, we will define it accordingly.

The presented study falls within the scope of NHESS. However, the paper is not ready for publication and needs major revisions before it would be acceptable for publication in the journal of NHESS.

Please also note the supplement to this comment:
https://www.nat-hazards-earth-syst-sci-discuss.net/nhess-2018-373/nhess-2018-373-AC1-supplement.pdf

---

## Author Comment (AC2) · 8 Apr 2019

This study proposed an improved method to calculate regional hydrological drought indices by incorporating auxiliary information, i.e., temperature, which is important to investigate hydrological extremes under global warming. However, the intro, results and discussion can be improved. I do have several concerns that should have been addressed before it can be considered for publication. 1. The authors used a new method to calculate hydrological drought index using temperature (LWSDI). LWSDI is not just a drought index, only when LWSDI is smaller than a threshold, it can be used for drought identification. Most importantly, the authors failed to demonstrate the improved

LWSDI is more appropriate than SPI/SPEI (or other hydrological drought indices) in investigating hydrological drought event. Although there are good relationships between LWSDI and SPE/SPEI, it is not convinced to prove LWSDI is better than the existing drought indices.

Authors Response: In the present work, we mainly focused on the improvement of time series data of precipitation records. The procedure of LWSDI is the same as of SPI and SPEI. However, LWSDI used improved time series data. The improvement of data is based on equation 1. In this equation, we used temperature as an auxiliary variable. Here, the Equation 1 is derived from sampling estimators. It is well-known method of statistics. Therefore, it's obvious that the better quality of data will produce better inference. Reference: Tarima, S., & Pavlov, D. (2006). Using auxiliary information in statistical function estimation. ESAIM: Probability and Statistics, 10, 11-23. Cochran, W. G. (2007). Sampling techniques. John Wiley & Sons.

2. The first two sentences of the ABSRACT and INTRODUCTION are the same. Please rewrite. Response: Thank you reviewer. In revised version of the manuscript, we have now rephrased the sentences accordingly. 3. Line 23-33: The authors failed to explain the relationships among different types of droughts."Hydrological drought occurs when dry weather patterns outweigh other climate conditions". This sentence is rather difficult to understand. Generally, hydrological drought is considered as the water shortage in surface/subsurface water during a certain period. The LWSDI is estimated by precipitation and temperature, which might be a kind of meteorological drought index. Response: This comment is related to the reviewer 2 comments. There we replied as, "In literature, there is too much confusion between the time scales and the definitions of drought such as meteorological, agricultural and hydrological (Bazrafshan et al., 2014). However, many authors have considered that the twelve-month time scale is suitable for defining hydrological drought (Svoboda et al., 2012). In the present work, a statistical methodology is adopted to improve time series data of precipitation. On the same manners of SPI, the proposed data is further used to compute drought

indices. The output of the proposed method of data is compatible to compare SPI-12 and SPEI-12. Therefore, we called LWSDI as a hydrological Index, because it gives monthly numerical standardized values at 12-month time scale. References: Bazraf-shan, J., Hejabi, S., & Rahimi, J. (2014). Drought monitoring using the multivariate standardized precipitation index (MSPI). Water resources management, 28(4), 1045-1060. Svoboda, M., Hayes, M., & Wood, D. (2012). Standardized precipitation index user guide. World Meteorological Organization Geneva, Switzerland." However, the sentence has now converted into a short paragraph which covers some definitions of droughts and literature about time scales.

4. Page 4 Line 40: 'various survey indicate that there is a positive correlation between rain and temperature. . .'.But in following, the authors indicate there are negative correlations between them (Rajeevan et al., 1998). It seems the method proposed by this study only can be used when there is a positive relationship between precipitation and temptation. Therefore, I strongly recommend the authors to give the relationship between them in the manuscript. Response: In casestudy experimentation, the correlation between precipitation and temperature is positive. For the revised paper, we have now prepared a table of correlation between precipitation and temperature. In addition, the significance of the correlation is tested by T-test. 5. Before selecting the appropriate probability distributions, I suggest the authors pay more attention to check whether the precipitation time series are stationary before statistical modelling. Response: We have now acknowledged by reviewer suggestions. We will make reanalysis accordingly. 6. The results and discussion are rather poor. The title of this study is "annual characterization. . .".The authors only show the statistics of the employed methods, but don't state the annual characterization of hydrological drought in Pakistan. There might be a severe drought in the study area during the study period, it would be better to identify and characterize these droughts, and compare with the other drought indices. Response: We will increase the quality of the result and discussion section. In addition, the grammar will be corrected from the native English speaker. The word annual characterization is used for the 12-month

time scale. In the revised version, we will focus on the 12-time scale. In addition, the categorical analysis will be performed for analyzing and comparing drought indices.

Please also note the supplement to this comment:
https://www.nat-hazards-earth-syst-sci-discuss.net/nhess-2018-373/nhess-2018-373-AC2-supplement.pdf

—————————————————————